# The evolution of RET inhibitor resistance in RET-driven lung and thyroid cancers

Ezra Y. Rosen[1,10], Helen H. Won[2,8,10], Youyun Zheng [3], Emiliano Cocco [4,5], Duygu Selcuklu[2], Yixiao Gong[2], Noah D. Friedman[4,6], Ino de Bruijn [2], Onur Sumer [2], Craig M. Bielski[4,6,7], Casey Savin[2], Caitlin Bourque[2], Christina Falcon[1], Nikeysha Clarke[1], Xiaohong Jing[2], Fanli Meng[2], Catherine Zimel[2], Sophie Shifman[3], Srushti Kittane[3], Fan Wu [3], Marc Ladanyi[3], Kevin Ebata[8], Jennifer Kherani[8], Barbara J. Brandhuber[8], James Fagin [1], Eric J. Sherman[1], Natasha Rekhtman[3], Michael F. Berger [2,3,7], Maurizio Scaltriti[4,9], David M. Hyman [8], Barry S. Taylor[2,4,6,7,8,10✉] & Alexander Drilon [1,10✉]

The efficacy of the highly selective RET inhibitor selpercatinib is now established in *RET*-driven cancers, and we sought to characterize the molecular determinants of response and resistance. We find that the pre-treatment genomic landscape does not shape the variability of treatment response except for rare instances of RAS-mediated primary resistance. By contrast, acquired selpercatinib resistance is driven by MAPK pathway reactivation by one of two distinct routes. In some patients, on- and off-target pathway reactivation via secondary *RET* solvent front mutations or *MET* amplifications are evident. In other patients, rare *RET*-wildtype tumor cell populations driven by an alternative mitogenic driver are selected for by treatment. Multiple distinct mechanisms are often observed in the same patient, suggesting polyclonal resistance may be common. Consequently, sequential RET-directed therapy may require combination treatment with inhibitors targeting alternative MAPK effectors, emphasizing the need for prospective characterization of selpercatinib-treated tumors at the time of monotherapy progression.

[1] Department of Medicine, Memorial Sloan Kettering Cancer Center, New York, NY, USA. [2] Marie-Josee and Henry R. Kravis Center for Molecular Oncology, Memorial Sloan Kettering Cancer Center, New York, NY, USA. [3] Department of Pathology, Memorial Sloan Kettering Cancer Center, New York, NY, USA. [4] Department of Human Oncology and Pathogenesis Program, Memorial Sloan Kettering Cancer Center, New York, NY, USA. [5] University of Miami, Miller School of Medicine, Department of Biochemistry and Molecular Biology/Sylvester Comprehensive Cancer Center, Miami, FL, USA. [6] Department of Epidemiology and Biostatistics, Memorial Sloan Kettering Cancer Center, New York, NY, USA. [7] Weill Cornell Medical College, New York, NY, USA. [8]Present address: Loxo Oncology at Lilly, Stamford, CT, USA. [9]Present address: AstraZeneca, Waltham, MA, USA. [10]These authors contributed equally: Ezra Y. Rosen, Helen H. Won, Barry S. Taylor, Alexander Drilon. ✉email: taylor.lab.msk@gmail.com; drilona@mskcc.org

Activating *RET* fusions or mutations drive oncogenic signaling in lung, thyroid, and other cancers. Oncogenic RET proteins occur via two primary mechanisms. First, gene fusions that contain the RET kinase domain produce constitutively active chimeric RET homodimers that drive cancer growth, and these fusions are most commonly present in non-small cell lung cancers and in papillary thyroid cancers. Second, somatic or germline point mutations in the kinase domain of RET result in aberrant RET kinase activation, causing medullary thyroid cancer.

The treatment of these patients with existing RET inhibitors (multikinase agents) yields only modest benefit and substantial toxicity. Selpercatinib is now established as one of two first-in-class FDA approved selective RET inhibitors for lung and thyroid cancers with RET mutations or fusions. Selpercatinib has shown unprecedented efficacy in RET-fusion-driven lung cancers with an objective response rate (ORR) of 64% (85% in previously untreated patients)[1], while in RET-altered thyroid cancer patients who had previously received cabozantinib, vandetanib, or both, the ORR was 69%[2]. While these clinical responses to selpercatinib are impressive, the differential response to selpercatinib remains unexplained. In addition, while in vitro work has been performed predicting activation of oncogenic pathways including RAS/MAPK signaling as a mechanism of resistance to selective RET inhibition[3], the precise description of these resistance mechanisms in treated patients remain largely unknown. Further, resistance to tyrosine kinase inhibitors is universal, therefore we anticipate emergence of resistance to selpercatinib in all treated patients.

In this work, we characterize the genomic determinants of response and resistance to selpercatinib. We find that the variability of treatment response is not affected by pre-treatment genomics, except in rare cases of RAS-mediated primary resistance. While diverse PI3K alterations can pre-exist and emerge on therapy, their temporal and response dynamics are inconsistent with therapeutic resistance. Conversely, MAPK pathway reactivation drives selpercatinib resistance either by (1) secondary *RET* solvent front mutations or *MET* amplifications, or (2) selection for *RET*-wild-type tumor cell populations characterized by an alternative driver. The emergence of these acquired MAPK pathway mutations suggests that later lines of therapy for RET-driven cancers may require strategies combining next-generation RET inhibitors with inhibitors targeting specific nodes in the MAPK pathway. Collectively, the pattern of resistance observed in *RET*-driven tumors demonstrates the intrinsic MAPK dependence of RET-altered cancers.

## Results

**Baseline patient attributes and response to selpercatinib.** To identify the determinants of response and resistance to RET inhibitor therapy in patients with *RET*-dependent cancers, we characterized pre-treatment and post-progression tumor biopsies and plasma cell-free DNA (cfDNA) specimens collected from 72 patients treated with the selective RET inhibitor selpercatinib on LIBRETTO-001, a registrational phase 1/2 trial (see "Methods", Supplemental Fig. 1). All patients had tumors harboring either a *RET* fusion, germline pathogenic variant, or somatic activating substitution or small in-frame insertion or deletion. Among *RET* fusion-positive cancers, lung adenocarcinomas predominated (81%), followed by papillary thyroid cancers (12%), and finally a variety of five other unique cancer types (Fig. 1a and Supplemental Table 1). Fusions involved a diversity of upstream partners and all retained the full RET kinase domain, with a subset also retaining regions of either the RET transmembrane or cysteine-rich domains (Fig. 1b). There was no statistically significant difference in treatment response as a function of the *RET* fusion partner. All patients with *RET* mutations had medullary thyroid

cancer (MTC). All somatic mutations were either statistically significant hotspot mutations identified from computational analysis of 31,447 sequenced cancers (see "Methods") or previously established oncogenic alleles[4]. All germline *RET* carriers had known pathogenic variants previously associated with increased heritable risk of medullary thyroid cancer (Fig. 1c).

In total, 19 patients received selpercatinib as their first line of therapy. Among the 53 remaining previously treated patients, 29 (55%) had received prior multi-targeted kinase inhibitor (MKI) therapy (Supplemental Tables 1 and 2). Selpercatinib efficacy was evident independent of either tissue of origin, alteration type (fusion versus mutation), or prior MKI exposure (Fig. 1d). The overall response rate according to RECIST version 1.1 per investigator assessment was 67% and 58% in the *RET* fusion-positive and *RET*-mutant cancers, respectively. Responses were durable, with 82% of patients ongoing on selpercatinib treatment at 1 year. At the time of data analysis, 47 patients remained on protocol therapy. These data are representative of the broader efficacy experience of selpercatinib which has been separately reported[1, 2, 5]. Given the similar treatment outcomes across tumor types and alteration classes, *RET*-mutant and fusion-positive patients were combined for all subsequent molecular analyses.

**Pre-treatment genomic profile and response to selpercatinib.** To assess whether the pre-treatment genomic profile of tumors shaped the initial sensitivity to selpercatinib, we performed targeted ($n = 51$) and/or whole-exome sequencing of tumor tissue ($n = 44$) acquired prior to treatment. The landscape of pre-existent genomic alterations overall reflected underlying tumor lineage. Among *RET*-mutant MTCs, five germline carriers were enrolled in addition to other somatic mutant cases, but an insufficient number of germline patients were treated to rigorously assess differences in response (Fig. 1d). In lung cancers harboring *RET* fusions, concurrent MAPK-activating alterations were not observed in tumor tissue, consistent with prior reports suggesting *RET* fusions are mutually exclusive with other mitogenic drivers[6]. Of note, patients with concurrent *TP53* mutations ($n = 8$, all with *RET* fusions) had shorter median PFS (HR = 3.5, 95% CI 1.3–9.7, $P = 0.016$; Supplemental Fig. 2a). Beyond *TP53*, there was no other pre-treatment co-mutation pattern associated with differential response to selpercatinib (Fig. 2a and Supplemental Fig. 3). Broader whole-exome sequencing of pre-treatment tumors in 44 patients indicated a landscape of mutational signatures consisting predominantly of age, APOBEC, MMR etiologies as well as a smoking signature in select *RET* fusion-positive lung cancers with a history of smoking, but was otherwise unremarkable (Supplemental Table 3). There was no apparent association between either tumor mutational burden or the degree of subclonal heterogeneity with the depth or durability of selpercatinib response (see "Methods", $P > 0.4$ for all comparisons; Supplemental Fig. 4a–c).

Pre-treatment circulating tumor-derived cell-free DNA (cfDNA) was sequenced in 68 of 72 cases using an ultra-deep and error-corrected sequencing assay targeting select exons of 129 key cancer genes. Plasma profiling detected the qualifying *RET* alteration in 95% (18/19) of *RET*-mutant cases (5 germline, 13 somatic) and in 74% (36/49) *RET* fusion-positive cases (Supplemental Fig. 5a). In five of the 14 patients in whom the *RET* alteration was not detected (four fusions and one mutation), no other mutations were detected at the limit of assay sensitivity (0.1%), potentially suggesting that a lack of shedding of tumor-derived DNA explains the discordance in these cases. In the other nine cases (all fusions), the mutant allele fraction of other detected mutations was low (0.1–0.8% allelic frequencies, 1–3 mutations per patient detected), suggesting that a minimal

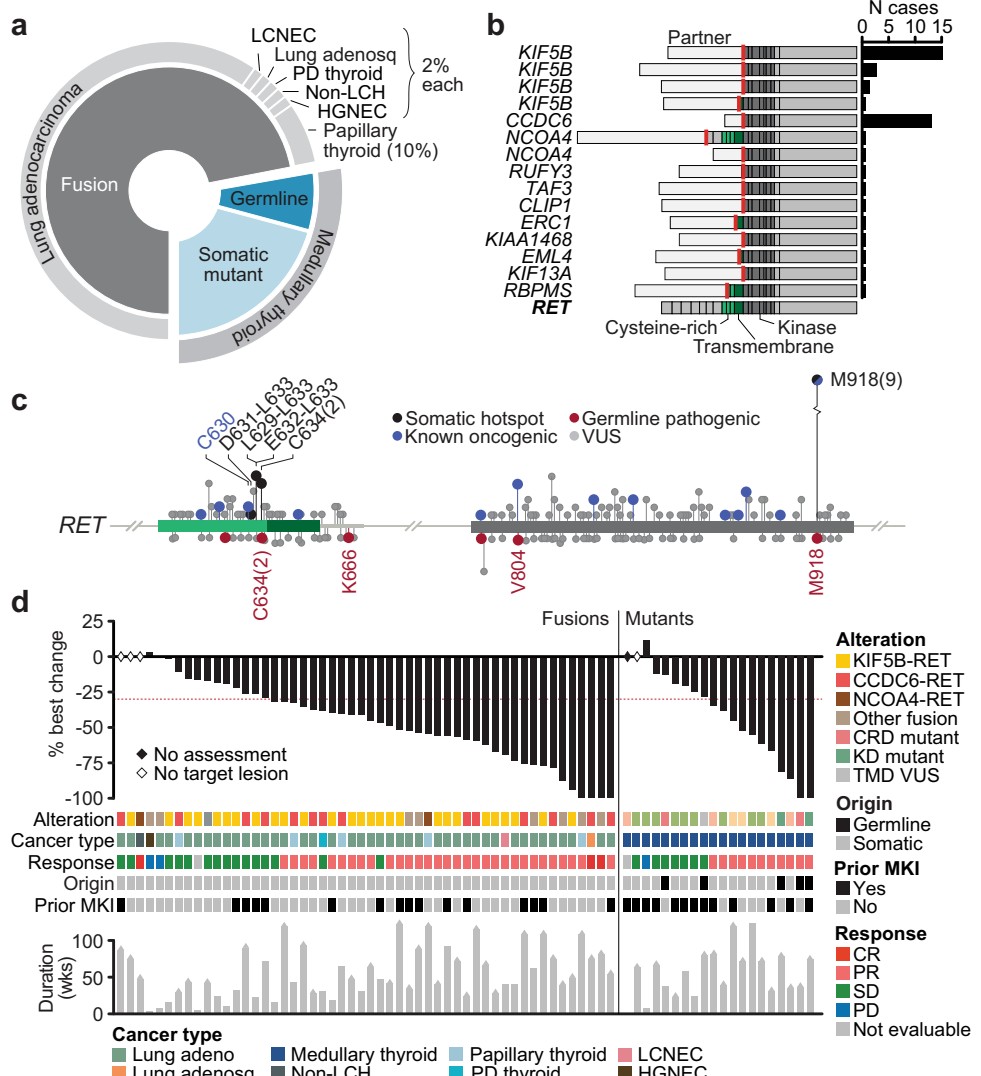

**Fig. 1 RET inhibition in RET-mutant solid cancers. a** The distribution of *RET* fusions and mutations (somatic or germline) in the study cohort by affected cancer type. PD, poorly differentiated; LCNEC, large cell neuroendocrine carcinoma of the lung; non-LCH, non-Langerhans cell histiocytosis of the skin; HGNEC, high-grade neuroendocrine carcinoma of the rectum. **b** The structure of all unique *RET* fusions in the study cohort (at right, number of affected cases). In light gray is the sequence from the indicated fusion partner. Red line demarcates fusion breakpoint. **c** Somatic mutations and germline variants (top and bottom, respectively) in two key regions of *RET* (left and right, respectively) in 31,447 prospectively sequenced human cancers. Labeled mutations correspond to enrolled patients. The mutational origin is indicated by the legend. Protein domains colored and indicated as in panel (**b**). **d** The clinical response of patients with *RET*-fusion or -mutant tumors to selpercatinib therapy is shown.

burden of tumor-derived cfDNA in circulation combined with potentially lower sensitivity for fusion detection explained observed discordance (Supplemental Fig. 5b, c, Supplemental Fig. 6). Notably, patients with no evidence or low rates of tumor shedding in baseline plasma presented with a lower burden of disease prior to treatment as measured by RECIST ($P = 0.02$, Supplemental Fig. 5d) and had superior progression-free survival on selpercatinib therapy (HR = 3.7, 95% CI 1.1–12.4, $P = 0.013$, likelihood ratio test; Fig. 2b). Among patients followed longitudinally by cfDNA ($n = 23$), the mutant allele frequency of the qualifying *RET* alteration (fusions or mutations) decreased in all but one evaluable patient (median 82.5% decrease, Fig. 2c). These data suggest that serial plasma ctDNA measurements can be used as a surrogate for molecular target engagement in treated patients.

**The impact of prior therapy and sensitivity to selpercatinib.** We next sought to determine the impact of prior therapy on the

initial sensitivity and molecular dynamics of selpercatinib treatment. In total, 29 patients were previously treated with MKIs vandetanib, cabozantinib, or RXDX-105. Prior exposure to these agents did not alter subsequent outcomes with selpercatinib (HR = 1.2, 95% CI 0.6–2.6, $P = 0.6$, Supplemental Fig. 2b). However, among patients previously treated with MKIs, baseline plasma profiling revealed coincident secondary *RET* mutations in addition to the primary documented *RET* driver that led to trial enrollment in four MTC patients (Supplemental Tables 2 and 4). In these 4 patients, the secondary *RET* mutations were all previously established activating mutations including V804M ($n = 2$) and G601E, K666E, and D898Y (1 each). Unlike the antecedent germline or somatic *RET* mutations that led to trial enrollment, all four activating alleles were detected only in plasma and at low frequencies (0.65–7%). Despite this prior therapy-mediated serial genetic evolution of *RET*, the plasma frequency of both the enrolling and secondary *RET* mutations decreased with selpercatinib treatment in all patients, indicating that the tumor cell

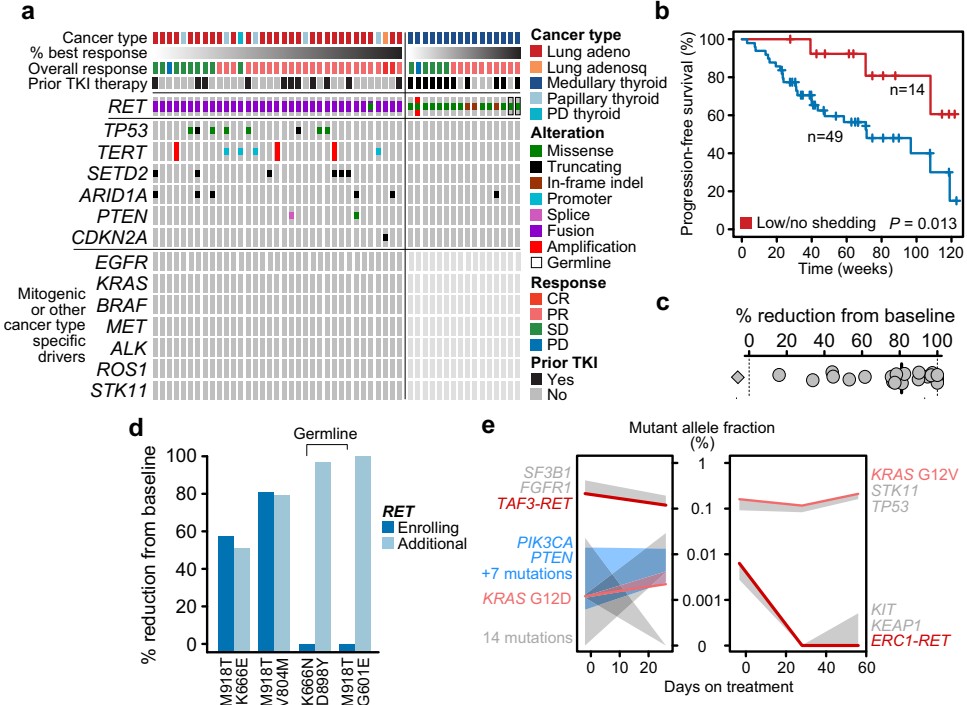

**Fig. 2 Determinants of initial sensitivity to RET inhibitor therapy. a** The genomic landscape of tumors acquired prior to selpercatinib treatment (dark and light gray, evaluated and no alterations observed). **b** Clinical benefit from selpercatinib treatment in patients with or without (no or minimal) evidence of tumor shedding in cfDNA (blue and red, respectively). *P* as indicated, likelihood ratio test. **c** The percent reduction in the mutant allele fraction of the enrolling *RET* alteration from baseline to the first time point after the start of treatment. Black line is the median (82.5%) reduction in all evaluable patients, the diamond is a single patient with an increased plasma frequency of *RET* after treatment initiation (patient had partial response lasting nearly a year, but second time point collected three weeks after progression). **d** In plasma, the percent reduction in the frequency of the *RET* enrolling alteration alongside the additional *RET* lesion present prior to selpercatinib therapy due to prior MKI therapy. **e** Two patients with primary resistance to selpercatinib in whom plasma cfDNA profiling at baseline identified *KRAS* mutations not detected by tumor tissue sequencing and presumed in distinct cancer populations (shading signifies the minimum and maximum allele frequency detected in each independent subclone) from the sensitizing *RET* fusion, which decreases upon treatment initiations.

with both *RET* alterations remained sensitive to RET inhibition (Fig. 2d). Among MKI-naive patients, only 1 of 19 had a secondary *RET* mutation, a T1078M variant of uncertain significance and presumed a passenger mutation.

**Mechanisms of primary resistance to selpercatinib**. As secondary *RET* mutations identified after prior MKI therapy did not preclude response, we sought to identify alternative potential mechanisms of primary resistance to selpercatinib. In total, 11 patients harbored PI3K pathway lesions including two patients for whom pre-treatment tumor tissue sequencing identified *PTEN* loss-of-function mutations (Fig. 2a), and nine patients for whom baseline plasma sequencing identified additional *PTEN* or *PIK3CA* mutations (Supplemental Fig. 7a). The patients harboring PI3K pathway alterations had a clinical benefit rate of 91% on selpercatinib (see "Methods"), indicating that these co-alterations did not preclude disease control, though longer clinical follow-up on larger groups of patients will be necessary to definitively establish the effect of PI3K alterations on efficacy (Supplemental Fig. 7a, b). Moreover, in three PI3K co-mutant patients for whom longitudinal cfDNA was profiled, the PI3K activating mutation decreased upon treatment initiation with a magnitude similar to that of the *RET* sensitizing alteration (Supplemental Fig. 7c). By contrast, in two patients with primary resistance to selpercatinib, pre-treatment plasma sequencing revealed *KRAS* mutations (G12D and G12V) shedding into circulation at low allele frequencies otherwise not detected by tumor tissue profiling, suggesting inter-tumoral heterogeneity and limitations of bulk tissue

sequencing (Fig. 2e). In both patients, the *RET* fusion decreased in frequency upon selpercatinib treatment, indicating therapeutic target engagement in that tumor cell population, with one patient indeed showing a mixed response on their first scan. Concurrently, the KRAS-mutant allele fraction conversely increased in allele frequency at the time of progression in both cases. While not definitively establishing the absence of a double-positive cell population, we confirmed reduced levels of inhibition as shown by higher levels of phosphorylated ERK in selpercatinib-treated cells co-expressing the observed *KRAS* alleles, (Supplemental Fig. 8), indicating the existence of distinct cancer cell populations from the sensitizing *RET* alteration. Collectively, these data suggest that multiple tumor clones with distinct mitogenic drivers (either *RET* or *KRAS*) may co-exist at baseline, each shedding into circulation at different rates, ultimately leading to either short duration of at-best mixed response or primary resistance (Fig. 2e). Therefore, despite there being no relationship between pre-treatment intratumoral heterogeneity assessed from tumor tissue and response to treatment, plasma sequencing, although inapt for the comprehensive landscape of evolutionary genomics, identified occult heterogeneity that was variably associated with treatment response and demonstrated the impact of the breadth and type of prospective clinical genomics to guide trial enrollment.

**Determinants of acquired resistance to selpercatinib**. Based in part on these instances of mechanism-based primary resistance, we sought to more broadly establish the determinants of acquired

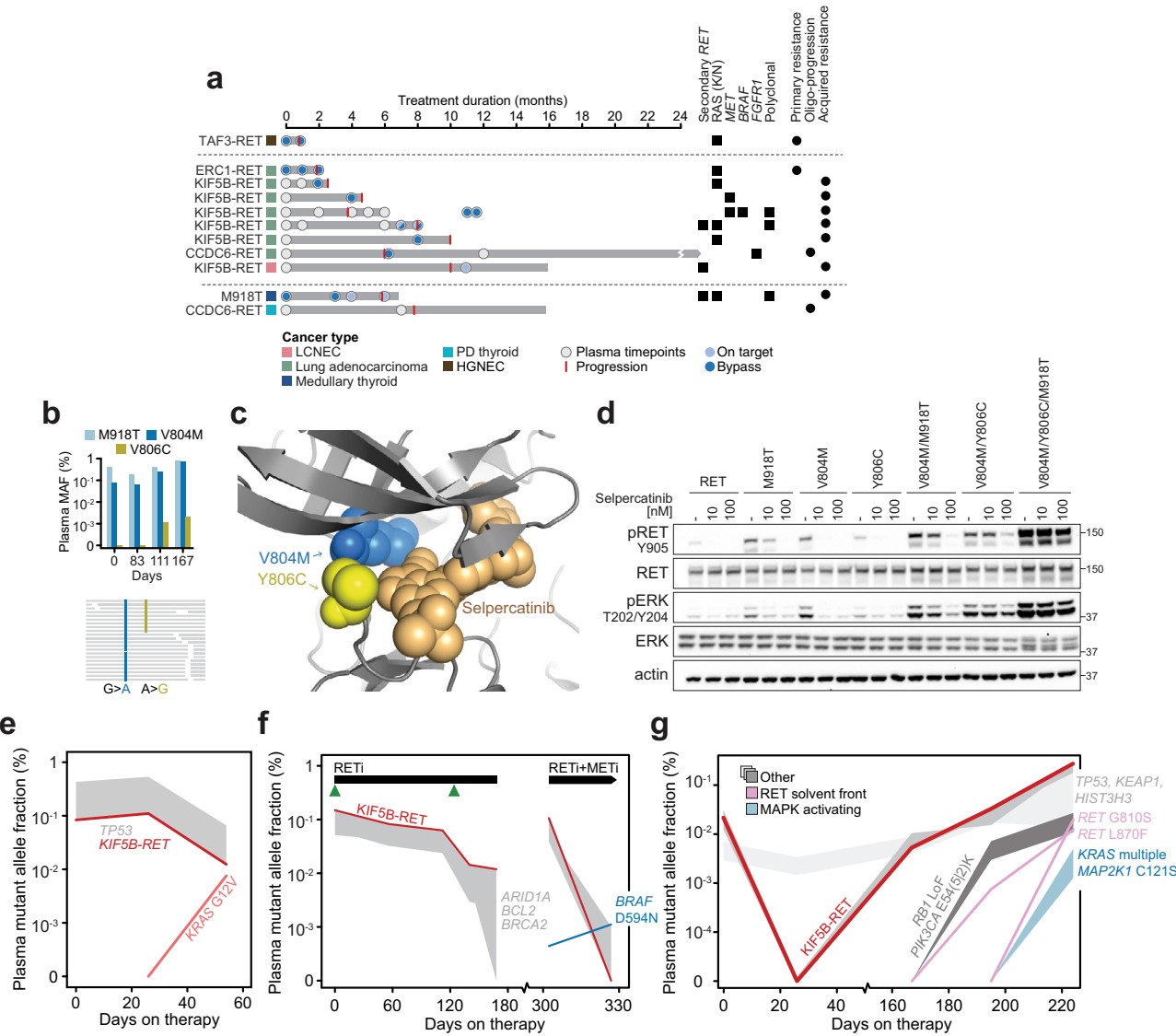

**Fig. 3 Mechanisms of RET inhibitor resistance. a** Selpercatinib-treated patients with either primary resistance, oligo-progression, or acquired resistance are shown indicating the duration of treatment response (arrows: ongoing) along with plasma sequencing time points and identified mechanism(s) of resistance (see legend). **b** Plasma *RET* M918T, V804M, and V806C mutant allele frequencies (top) and sequencing reads reflecting *cis*-acting mutants at progression timepoint (bottom). **c** The structure of *RET* bound by selpercatinib (as indicated) with *cis* V804M and Y806C composite mutations. **d** In 293T cells, expressing double in *cis* mutants (*RET* V804M/Y806C) reduced sensitivity to selpercatinib as measured by the dose-dependent decrease of both pRET and pERK by western blot. The triple *cis* mutant *RET* M918T/V804M/Y806C was resistant to selpercatinib (*n* = 2 experiments). **e** Longitudinal plasma cfDNA profiling indicates the *RET* fusion-positive and *PIK3CA*-mutant clone responding with a likely *RET*-wild-type *KRAS* G12V subclone emerging at resistance (independent cellular populations represented by shading which signifies the minimum and maximum allele frequency detected in each independent subclone). **f** As in panel (**e**) but for a patient with a subclonal *MET* amplification (green) that is selected for by selpercatinib therapy despite reduction of the *RET* fusion in plasma. Subsequent combined *MET* and *RET* inhibition selects for the outgrowth of a *BRAF* D594N-mutant subclone. **g** As in panel (**e**) but for a patient with a complex pattern of polyclonal resistance with cell populations emerging mediated by distinct and non-overlapping on-target and bypass mechanisms of selpercatinib resistance.

resistance to selpercatinib. In 18 of 27 patients who had progressed on selpercatinib, adequate pre- and post-progression tumor or longitudinal plasma specimens existed for sequencing (see "Methods", Supplemental Fig. 1, Supplemental Table 5). Overall, we identified a genetically driven mechanism of resistance in 11 of 18 evaluable progressing patients. Three patients developed on-target resistance with the emergence of secondary *RET* mutations, of which multiple were identified in individual subclones in the same patient (Fig. 3a). Specifically, two patients with truncal *KIF5B-RET* fusions developed *RET* solvent front mutations (G810C or G810S), consistent with our prior case series demonstrating bulky substitution at this residue leads to

steric hindrance with available selective RET inhibitors[7]. The third *RET* M918T-mutant medullary thyroid cancer patient in whom we detected a gatekeeper V804M mutation at baseline subsequently developed an acquired Y806C mutation in cis (Fig. 3b). Selpercatinib binds the pocket adjacent to V804 and Y806 and is thought to maintain binding activity against either mutation alone. We hypothesized that in tumor cells with the *cis* composite mutation of both residues, however, selpercatinib binding is unaccommodated by the altered pocket (Fig. 3c). To test this hypothesis, we expressed each of these *RET* mutations alone or in combination in 293T cells and assessed the dose-dependent effect of selpercatinib on signaling. While expressing

double in cis RET mutants reduced sensitivity to lower concentrations of selpercatinib, the compound triple mutant (M918T/V804M/Y806C) that emerged on therapy abrogated the effect of selpercatinib on phosphorylated RET and ERK (Fig. 3d). These in vitro experimental, structural, genomic, and clinical data indicate that the serial genetic evolution of mutant RET can lead to selpercatinib resistance in patients. This emergence of multiple RET mutations expands on a pattern recently observed in ALK-driven lung cancers with sequentially more potent ALK inhibitors, whereby compound mutations confer a similar response liability[7, 8].

By contrast, emergent MAPK-activating alterations similar to those mediating primary resistance indicated that bypass signaling was another important mechanism of selpercatinib resistance. Overall, seven patients acquired KRAS (G12A/R/V, G13D, A59del), NRAS (G13D, Q61R), or BRAF activating mutations or MET or FGFR1 amplifications (Fig. 3a, Supplemental Table 5). Typical among these was a KIF5B-RET fusion-positive lung cancer patient for whom a KRAS G12V mutation was not detected in either baseline tissue or plasma but emerged on therapy (Fig. 3e). Notably, the frequency of the RET fusion in plasma at progression was still 85% lower than the pre-treatment baseline while the KRAS mutation was simultaneously growing out. This suggests that the RET fusion and KRAS mutation were likely not present in the same cell population and selpercatinib created a selective pressure that eliminated the still-sensitive RET fusion-positive cells but fostered the resistant KRAS-mutant cell population (Fig. 3e). This pattern is consistent with that observed in the few patients to exhibit primary resistance (Fig. 2d).

Beyond mutant RAS, MET amplifications were observed in multiple progressing patients after only modest and short-lived responses to selpercatinib (PFS of less than or equal to 6 months), consistent with prior reports[9, 10]. In one patient, pre-treatment tissue sequencing revealed a subclonal MET amplification which grew out upon selpercatinib treatment. Testing the clinical hypothesis that combination therapy may re-sensitize this patient, they were subsequently treated with and re-responded to the combination of selpercatinib and the MET inhibitor crizotinib[11]. Treatment was discontinued after four months due to an unrelated adverse event. However, plasma sequencing during combination therapy revealed an emerging BRAF D594N mutation, a functionally distinct kinase-dead RAS- and BRAF dimer-dependent mutant[12] that reinforces the RAS/MAPK dependence of RET-driven tumors (Fig. 3f). Two other patients with an extensive burden of disease at trial enrollment had oligo-progression on selpercatinib with ongoing clinical benefit. In one such patient, tissue sequencing revealed a focal FGFR1 amplification was acquired in the solitary oligo-progressing site that was absent prior to selpercatinib treatment and may also mediate MAPK reactivation[13]. After receiving local radiation therapy, both patients continued on selpercatinib, ultimately remaining on therapy for two and four times longer than their initial response, respectively.

This diversity of MAPK-driven mechanisms of primary or acquired resistance was also evident within individual patients, indicating a complex pattern of polyclonal resistance to selpercatinib therapy can emerge. Typifying this pattern were two patients in whom solvent-front RET mutations co-existed with KRAS G12A and G12R mutations. Plasma sequencing indicated that these were likely distinct resistant subclones as they emerged at different time points (Fig. 3g). Presumably, the solvent front mutation was acquired in the RET fusion-positive clone, which was distinct from the likely RET-wild-type cell population that was driven by KRAS G12, as was the case for patients with RAS-driven primary resistance (Fig. 2d).

While these analyses identified a mechanism of resistance in 11 of 18 evaluable patients with disease progression (61%), our data do not preclude a non-genetic mechanism of resistance in the remaining cases. Notably, beyond the acquisition of these discrete resistance mechanisms, in four cases with matched pre- and post-treatment tissue samples, we did not observe significant clonal evolution. Nevertheless, to examine the possibility of a mechanism of resistance beyond discrete alterations leading to either on-target or bypass resistance, we explored lineage plasticity which has been associated with resistance to EGFR inhibitors such as osimertinib in NSCLC[14]. We found no evidence of lineage/histology changes as a source of acquired resistance in evaluable patients (three RET fusion-positive lung cancers) who developed drug resistance without an identified clear genetic driver where a post-progression biopsy was available (Supplemental Fig. 9).

## Discussion

Collectively, these results indicate that key mechanisms of primary and acquired resistance to selective RET inhibitor therapy in RET-altered lung and thyroid cancers converge on MAPK pathway activation, with diverse yet partially overlapping and often co-existent mechanisms in individual patients. In many cases, these MAPK pathway-activating events can be polyclonal within the same patient. Moreover, these events can occur alone or in combination with on-target RET resistance mutations, of which solvent front mutations appear the most common. Similar patterns have been observed in other kinases involved in fusions in cancer such as ALK, ROS, and TRK[15–17]. Typified by selpercatinib in mutant RET tumors, a newer generation of superior inhibitors targeting mutant oncogenes are beginning to challenge long-held assumptions about the perceived mutual exclusivity of mitogenic drivers in lung cancer and beyond, uncovering co-existent mitogenic drivers whereby one ostensibly fitter sensitizing allele becomes dominant after which therapy reveals the other as a mechanism of escape. Ultimately, identifying the mechanisms of primary and acquired resistance to selective inhibitors in oncogene-addicted tumors has facilitated the development of multiple therapeutic strategies that can dramatically extend clinical benefit[18, 19]. Our results suggest that the strategy of sequential TKI treatment developed for other oncogene-driven cancers may not be as broadly effective in RET-driven cancers, possibly due to the selectivity and potency of currently available purpose-built RET inhibitors. Alternative strategies may be necessary to extend the profound clinical benefit of selpercatinib therapy.

## Methods

**Study cohort**. All patients had a confirmed RET activating fusion or mutation, qualified for treatment based on screening guidelines, and were subsequently treated with selpercatinib on the phase 1/2 LIBRETTO-001 trial at Memorial Sloan Kettering Cancer Center (ClinicalTrials.gov #NCT03157128). This protocol was approved by the Institutional Review Board at Memorial Sloan Kettering Cancer Center. Selpercatinib was dosed as specified by the protocol in the clinical manuscript describing the efficacy of this drug[1]. One patient in the study cohort was treated on a single patient-use protocol due to the presence of leptomeningeal disease. All patients met the previously described eligibility criteria[20, 21], having adequate hematopoietic, hepatic, kidney, and cardiac function for inclusion. All patients provided written informed consent including for clinical and research next-generation sequencing. Included in the study cohort are patients enrolled between May 2017 and clinical data freeze for analysis in December 2019. Disease response was assessed using RECIST 1.1 criteria. Clinical benefit rate (CBR) was defined as the sum of best overall response rates of SD, PR, and CR.

A total of 70 patient samples (two patients were excluded as lost to follow-up) were included for correlative genomic analyses (51 fusion, 19 mutations) from the December 2019 clinical data freeze. Specimens underwent either tumor tissue sequencing using the MSK-IMPACT targeted sequencing assay (up to 468 cancer-associated genes and regions) or whole-exome sequencing (WES) as well as plasma cfDNA sequencing using the MSK-ACCESS targeted sequencing panel (see "Plasma cell-free DNA sequencing" for details) of which 47 patients were still receiving ongoing treatment. For tumor tissue sequencing, 19 patients with insufficient tissue or quality were excluded. A total of 51 pre-treatment tumor tissue samples and their matched germline controls were sequenced using MSK-

IMPACT, of which samples from 44 patients also underwent further WES. For plasma cfDNA sequencing, a total of 68 pre-treatment baseline samples were sequenced using MSK-ACCESS. Of the 70 patients, 28 patients had sufficient samples for pre- and post-treatment sequencing. This included nine patients with pre- and post-treatment tumor tissue sequencing with MSK-IMPACT and 26 patients with longitudinal cfDNA time points for sequencing with MSK-ACCESS. Longitudinal plasma sequencing included 18 patients who clinically progressed on selpercatinib and an additional eight MTC patients who received prior MKI therapy (Supplemental Fig. 1).

**Tumor tissue sequencing and analyses**. Genomic DNA from pre-treatment tumor tissue and matched germline DNA were sequenced for 51 patients using MSK-IMPACT to identify somatic single-nucleotide variants, small insertions and deletions, copy-number alterations, and structural variants as previously described[22]. To enhance the breadth of somatic alterations (mutational signatures, DNA copy number, and tumor mutational burden), broader WES was also performed on the existing MSK-IMPACT barcoded libraries in 44 of these 51 patients using the IDT xGen Exome Research Panel (v1.0) according to the manufacturer's protocol to achieve an average coverage of 208X and 89X for the tumor and matched normal specimens, respectively. WES data were processed and analyzed using the Tempo pipeline (v1.3, https://ccstempo.netlify.app/). Briefly, demultiplexed FASTQ files were aligned to the b37 assembly of the human reference genome from the GATK bundle using BWA mem (v0.7.17). Aligned reads were converted and sorted into BAM files using samtools (v1.9) and marked for PCR duplicates using GATK MarkDuplicates (v3.8-1). Somatic mutations (SNVs and small indels) were called in tumor-normal pairs using MuTect2 (v4.1.0.0) and Strelka2 (v2.9.10), and structural variants were detected using Delly (v0.8.2) and Manta (v1.5.0). Variants were annotated and filtered for recurrent artifacts and false positives using methods as previously described[23].

Sequencing data were used to confirm the enrolling *RET* alteration; define co-mutational patterns and tumor mutational burden (TMB); assess microsatellite instability (MSI) status; and infer allele-specific copy number, clonality, and mutational signature decomposition, all using previously described methods. Briefly, for patients with WES data, TMB was calculated as the number of non-synonymous mutations per targeted megabase, and MSI status was assessed using MSIsensor (v0.5)[24]. MSIsensor scores <10 were classified as microsatellite stable and >10 were classified as MSI-high using a previously validated cut-off score[25]. We estimated tumor purity and ploidy as well as total and allele-specific copy number using the FACETS algorithm (v0.5.14, http://github.com/mskcc/facets)[26]. To infer clonality and subclonal heterogeneity, cancer cell fractions (CCFs) were estimated with 95% confidence intervals by integrating somatic mutations with FACETS-derived joint segmentation[27, 28]. Briefly, CCFs were calculated for all mutations using the mutant allele fraction, locus-specific sequencing coverage, and the tumor purity estimate using a binomial distribution and maximum likelihood estimation to generate posterior probabilities[27, 29]. Mutations were classified as either clonal or subclonal on the following criteria: clonal if the upper bound of the 95% confidence interval was >0.85, otherwise subclonal. Tumors were classified as having significant subclonal heterogeneity if more than 30% of the somatic non-synonymous mutations were subclonal per the aforementioned criteria. In addition, in samples with five or more SNVs, mutations were decomposed into one of ~30 constituent mutational signatures (http://github.com/mskcc/mutation-signatures) and annotated for their presumed etiology as previously described[30].

*RET* mutational hotspots were identified using a previously described method[31] applied to an extended cohort of 31,447 prospectively sequenced human cancers using MSK-IMPACT. Briefly, the statistical significance of substitution hotspots was assessed with a truncated binomial probability model incorporating underlying features of gene-specific mutation rates including gene length, gene- and position-specific mutability, and overall mutational burden. Hotspots with a false discovery rate (Benjamini and Yekutieli method) adjusted *p*-value <0.1 were considered significant. For small in-frame indel hotspots, indels were grouped using a maximal common region approach to count the number of indel events spanning a particular region and the same binomial probability model was applied without a background model (http://github.com/taylor-lab/hotspots). In addition, alterations were annotated as oncogenic using the OncoKB knowledgebase[4].

**Plasma cell-free DNA sequencing**. Cell-free DNA (4–10 ng) and white blood cells (200 ng) were extracted from all plasma time points (baseline, *n* = 68; on- and post-treatment, *n* = 26; matched buffy-coat, *n* = 9) and sequenced using a custom, ultra-deep coverage sequencing panel (MSK-ACCESS) that targets key exons of 129 genes and introns of 10 genes harboring recurrent breakpoints[32]. Additional probes targeting all exons and introns of *RET* were spiked in to increase the coverage and sensitivity of *RET* mutation and fusion detection. Briefly, this custom assay utilizes duplex unique molecular identifiers (UMIs) and dual index barcodes to minimize background sequencing errors to generate error-suppressed consensus reads with representation from both strands of the original cfDNA duplex. Briefly, FASTQ files were processed using a custom pipeline that trims the UMIs, aligns the sequences to the human genome, and collapses PCR replicates into consensus sequences using the in-house Marianas algorithm (https://github.com/mskcc/Marianas). Consensus reads with representation from both strands of the original

cfDNA duplex were used for de novo variant calling using VarDict (v1.5.1). Synonymous, intronic, or intergenic mutations were excluded from further analysis and all exonic mutation calling required at least one consensus read at a known cancer hotspot or at least three consensus reads at non-hotspot sites and were called down to the limit-of-detection of 0.1% allele frequency[17]. All cfDNA samples were sequenced to an average depth of approximately 19,348-fold (range: 11,139–32,848) total coverage and 1271-fold (range: 552–3134) unique duplex coverage. In samples with matched buffy-coat, germline and potential clonal hematopoiesis (CH) variants were removed from analysis if the mutation was observed in the buffy-coat normal. Otherwise, variants were called against unmatched healthy plasma donors to identify any specimen type-related artifacts. To further exclude the possibility that low mutant allele frequency somatic mutations observed in plasma originated from CH rather than the tumor, we assessed the fragment size distribution of the variants. We observed shorter fragment sizes for tumor-derived variants when compared to germline variants originating from non-cancer cells (white blood cells, hematopoietic cells), as previously reported[33]. De novo *RET* fusion calling was performed using Manta (v1.5.0) and were manually reviewed and additionally confirmed using a priori fusion breakpoint data from tumor tissue sequencing. Fusion allele frequencies were quantified by calculating the depth of coverage from both sides of the breakpoint. Cell-free DNA samples with the detection of either the enrolling *RET* alteration or other somatic alterations were considered to be shedding ctDNA in plasma. *RET* alterations in a subset of baseline plasma samples (*n* = 49) were validated using an orthogonal commercial assay targeting key regions of 73 cancer-related genes (Guardant360) using previously published methods[34]. All relevant findings were cross-validated and confirmed.

**Outcome analyses**. Clinical benefit was defined as stable disease (SD), partial response (PR), or complete response (CR) lasting 24 weeks or greater, or these same response types in patients if they were continuing on treatment but their total duration of treatment was <24 weeks at time of data freeze. Two patients withdrew consent prior to confirmation of response on imaging assessment and were thus removed from outcome analysis. We defined progression of disease (PD) as the date of radiologic progression or death. We calculated confidence intervals for response rate using the Clopper-Pearson method, and used the likelihood ratio test for assessing the significance between groups for all outcome analyses. All genomic correlative associated *P* values were generated using one-way ANOVA or Wilcoxon test accordingly.

**Plasmid generation, cell culture, western blots, and immunohistochemistry**. The pDONR RET plasmid was purchased from ADDGENE (# 23906). Mutagenesis was performed on this vector to generate the RET V804M single mutant: RETV804MmutF: CTCCTCATCaTGGAGTACGCC and RETV804MmutR: GGCGTACTCCATgATGATGAGGAG; the RET Y806C single mutant: RETY806CmutF: ATCGTGGAGTgCGCCAAATAC and RETY806CmutR: GTATTTGGCGcACTCCACGAT; the RET M918T single mutant: RETM918TmutF: GTTAAATGGAcGGCAATTGAA and RETM918TmutR: TTCAATTGCCgTCCATTTAAC; the RET V804M/Y806C double mutant: RETV804MY806CmutF: ATCaTGGAGTgCGCCAAATAC and RETV804MY806CmutR. To generate the RET V804M/M918T double mutant the M918T primers pairs were used on the RET V804M single mutant. The same set of primers was used on the RET V804M/M918T to generate the triple mutant. Gateway™ LR Clonase™ II Enzyme mix (Invitrogen; #11791020) was used to clone the above from pDONR into the PLX302 destination vector. The PLX302 KRAS, KRAS G12A, and G12D were generated as previously described[16]. The RET encoding constructs were used to transiently transfect 293T cells while the KRAS constructs to transfect MZ-CRC-1 Medullary Thyroid Carcinoma cells obtained from Dr. Fagin's laboratory using Lipofectamine™ 3000 Transfection Reagent (Thermo Fisher; #L30000008) according to the manufacturer's protocol. Twenty-four hours after transfection, cells were treated with the indicated concentrations of selpercatinib for 30 min. After incubation, cells were frozen. The day after protein lysates were extracted, quantified and used for western blots. The following antibodies were used: RET mAB (1:1000 dilution, CST, #14556), pRET Y905 (1:500 dilution, CST, #3221), ERK (1:1000 dilution, CST, #9102S), pERK T202/Y204 (1:1000 dilution, CST, #4370S), KRAS (1:500 dilution, LSBio, #LS-C175665-100) and actin (1:2000 dilution, CST, #4970S). Western blot experiments shown are representative of two biological replicates. Immunohistochemistry (IHC) antibodies were used as follows: CD56 (1:100 dilution, clone MRQ42, Cell Marque), Chromagrannin A (1:4 dilution, clone LK2H10, Ventana), Ki-67 (1:1000 dilution, clone MIB1, Dako), Synaptophysin (1:2000 dilution, clone SNP88, Bio Genex). Due to limited tissue availability this IHC was performed on single tissue sections.

**Reporting summary**. Further information on research design is available in the Nature Research Reporting Summary linked to this article.

## Data availability

All patient-level clinical and genomic data from the study cohort is available through the cBioPortal for Cancer Genomics[35, 36]: https://cbioportal.mskcc.org/study/summary?id=mixed_selpercatinib_2020. The whole-exome sequencing data are deposited at

dbGAP under the following URL: https://www.ncbi.nlm.nih.gov/projects/gap/cgi-bin/study.cgi?study_id=phs001783.v2.p1. Source data are provided with this paper.

## Code availability

All other genomic and clinical data accompanies the manuscript and is available as Supplemental Data and Supplementary Information. All statistical analysis and figures were generated using R software. Source code for these analyses is available at http://github.com/taylor-lab/RET.

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

## Acknowledgements

This work was supported by National Institutes of Health awards P30 CA008748, T32 CA009512-29A1 (E.Y.R.), T32 CA160001-08 (E.C.), R01 CA207244 (B.S.T.), R01 CA204749 (B.S.T.), R01 CA245069 (B.S.T.), and R01 CA251591 (A.D.). This work was also supported by an MSK scholar prize (E.C.).

## Author contributions

E.Y.R., B.S.T., and A.D. conceived the study. E.Y.R., H.H.W., Y.Z., E.C., Y.G., N.D.F., I.B., O.S., C.M.B., S.S., B.J.B., and B.S.T. designed and performed the analysis. D.S., C.S., C.B., C.F., N.C., X.J., F.M., C.Z., S.K., F.W., M.L., K.E., J.K., J.F., E.J.S., N.R., M.F.B., M.S., D.M.H., and A.D. assisted with genomic and clinical data collection. E.Y.R., H.H.W., B.S.T., and A.D. wrote the manuscript with input from all authors.

## Competing interests

E.Y.R. received research funding from Bayer. M.F.B. received honoraria from Roche and research funding from Grail outside of this work. He is also a co-inventor on a provisional patent application for systems and methods for detecting cancer via cfDNA screening. A.D. received honoraria for advisory board activities for Ignyta/Genentech/Roche, Loxo/Bayer/Eli Lilly, Takeda/Ariad/Millenium, TP Therapeutics, AstraZeneca, Pfizer, Blueprint Medicines, Helsinn, Beigene, BergenBio, Hengrui Therapeutics, Exelixis, Tyra Biosciences, Verastem, MORE Health, Abbvie, 14ner/Elevation Oncology, Remedica Ltd., ArcherDX, Monopteros; and research funding from Pfizer, Exelixis, GlaxoSmithKlein, Teva, Taiho, PharmaMar, Foundation Medicine; royalties from Wolters Kluwer; and miscellaneous expenses from Merck, Puma, Merus, and Boehringer Ingelheim. M.S. is an employee of AstraZeneca. D.M.H. is an employee of Loxo Oncology, a wholly-owned subsidiary of Eli Lilly. H.H.W. and B.S.T. are currently employees of Loxo Oncology, a wholly-owned subsidiary of Eli Lilly. B.S.T. reports receiving honoraria and research funding from Genentech and advisory board activities for Boehringer Ingelheim. All stated activities were outside of the work described herein. No other disclosures were noted.
