## [Peer Review File · Nature Communications]

Reviewers' Comments:

Reviewer #1:

Remarks to the Author:

This report details mechanisms of resistance from RET+ (fusion or mutation) primarily from NSCLC and MTC following treatment on selpercatinib on the phase 1/2 LIBRETTO-001 clinical trial. The uniformity of treatment on a standardized clinical trial protocol is a rarity in series that describe resistance. All patients were from a single site (MSKCC), which is only a minor criticism. The authors report both on ctDNA and tumor-based sequencing results which adds to the strength of results. They also report on both intrinsic and acquired resistance. As with most of these series, a large percentage of patients have no detectable or identifiable mechanism of resistance, which remains an important unmet need. All of the mechanisms of resistance identified in this report have been reported previously for RET or other oncogenes including ALK, ROS1, EGFR and NTRK. The authors have adequately cited prior reports of MET-mediated resistance (although not the publication by Lin et al. *Annals of Oncology* 2020, which described a variety of resistance mechanisms to both selpercatinib and pralsetinib), but have not cited in vitro work that predicted NRAS mutations as a mechanism of resistance to RET inhibitors (Nelson-Taylor et al., *Mol Can Ther* 2017).

The PIK3CA mutation data suggests a trend towards worse outcome (also made stronger by author response showing lower mutation rate and shallower depth of response). The lack of statistical significance is likely driven by small patient numbers, thus the authors should acknowledge that pre-existing PIK3CA mutations may impact outcomes.

Reviewer 1:

This report details mechanisms of resistance from RET+ (fusion or mutation) primarily from NSCLC and MTC following treatment on selpercatinib on the phase 1/2 LIBRETTO-001 clinical trial. The uniformity of treatment on a standardized clinical trial protocol is a rarity in series that describe resistance. All patients were from a single site (MSKCC), which is only a minor criticism. The authors report both on ctDNA and tumor-based sequencing results which adds to the strength of results. They also report on both intrinsic and acquired resistance. As with most of these series, a large percentage of patients have no detectable or identifiable mechanism of resistance, which remains an important unmet need. All of the mechanisms of resistance identified in this report have been reported previously for RET or other oncogenes including ALK, ROS1, EGFR and NTRK.

Question 1:

The authors have adequately cited prior reports of MET-mediated resistance (although not the publication by Lin et al.

Annals of Oncology 2020, which described a variety of resistance mechanisms to both selpercatinib and pralsetinib), but have not cited in vitro work that predicted NRAS mutations as a mechanism of resistance to RET inhibitors (Nelson-Taylor et al., Mol Can Ther 2017).

Answer 1:

We thank the reviewer for bringing this to our attention, and we now cite these two papers in the manuscript file, and we now remark in the manuscript on the in vitro work that predicted RAS/MAPK mutations as a mechanism of resistance to RET inhibitors.

Question 2:

The PIK3CA mutation data suggests a trend towards worse outcome (also made stronger by author response showing lower mutation rate and shallower depth of response). The lack of statistical significance is likely driven by small patient numbers, thus the authors should acknowledge that pre-existing PIK3CA mutations may impact outcomes.

Answer 2:

We thank the reviewer for this comment and we agree that small patient numbers may be driving the lack of statistical significance here. We have therefore modified the text in the discussion to explain that more work on larger groups of patients followed for longer duration will be needed to firmly establish the effect of PI3K mutation on the outcomes of these patients.